# Mapping molecular assemblies with fluorescence microscopy and object-based spatial statistics

Thibault Lagache [1,5], Alexandre Grassart[2], Stéphane Dallongeville[1], Orestis Faklaris[3], Nathalie Sauvonnet[2], Alexandre Dufour[1], Lydia Danglot [4] & Jean-Christophe Olivo-Marin [1]

Elucidating protein functions and molecular organisation requires to localise precisely single or aggregated molecules and analyse their spatial distributions. We develop a statistical method SODA (Statistical Object Distance Analysis) that uses either micro- or nanoscopy to significantly improve on standard co-localisation techniques. Our method considers cellular geometry and densities of molecules to provide statistical maps of isolated and associated (coupled) molecules. We use SODA with three-colour structured-illumination microscopy (SIM) images of hippocampal neurons, and statistically characterise spatial organisation of thousands of synapses. We show that presynaptic synapsin is arranged in asymmetric triangle with the 2 postsynaptic markers homer and PSD95, indicating a deeper localisation of homer. We then determine stoichiometry and distance between localisations of two synaptic vesicle proteins with 3D-STORM. These findings give insights into the protein organisation at the synapse, and prove the efficiency of SODA to quantitatively assess the geometry of molecular assemblies.

[1] Institut Pasteur, BioImage Analysis Unit. CNRS UMR 3691. 25 rue du Docteur Roux, 75724 Paris Cedex 15, France. [2] Institut Pasteur, Molecular Microbial Pathogenesis Unit. INSERM U1202. 28 rue du Docteur Roux, 75724 Paris Cedex 15, France. [3] CNRS UMR7592, Institut Jacques Monod, Université Paris Diderot, 15 rue Hélène Brion, 75013 Paris, France. [4] Inserm U894 Center for Psychiatry and Neuroscience, Team Membrane traffic in healthy and diseased brain, 102-108 rue de la Santé, 75014 Paris, France. [5] Present address: Department of Biological Sciences, Columbia University, New York NY USA. Correspondence and requests for materials should be addressed to L.D. (email: lydia.danglot@inserm.fr) or to J.-C.O-M. (email: jcolivo@pasteur.fr)

To understand protein functions and molecular network organisation, it is important to determine the localisation of proteins at subcellular level. The in cellulo probing of molecular assemblies can be performed with optics-based techniques like fluorescence cross correlation spectroscopy[1] or Forster Resonance Energy Transfer[2] or, more commonly, with microscopy techniques which visualise the localisations of molecules. The analysis of the spatial proximity between different molecules of interest is traditionally achieved by labelling proteins with different fluorophores (in general red and green fluorophores), and quantifying their spatial co-localisation[3,4]. Three main issues impede the robustness of co-localisation analysis and have led to several technical developments over the years. First, co-localisation analysis is sensitive to image noise and background intensity[5] (Fig. 1). Thus, image denoising is a unavoidable pre-requisite to any co-localisation analysis, and many algorithms from signal thresholding[6], to energy minimisation[7,8] and wavelet-based detection of spots[9,10] have been introduced and used so far in co-localisation analysis. Second, traditional co-localisation index are based on signal correlation (e.g. Pearson Correlation Coefficient (PCC)[4,11]) and overlap (e.g., Manders Overlap Coefficient (MOC)[3,6,12]) and thus depend on the size of the point spread function (PSF) of the microscope (see also Table 1 for a more complete list of correlation and overlap methods). Moreover, these methods cannot be used for single-molecule localisation microscopy where the nanometre scale localisation of molecules is directly estimated from the sequential activation of blinking molecules[13,14]. Thus, distance-based methods that evaluate the spatial association (coupling) between objects (spots or localisations) based on their relative positions rather than their fluorescence correlation or overlap have been introduced[10,15,16]. Yet, while these distance-based methods have demonstrated their statistical power in detecting coupling between objects, they do not clearly measure the number of objects' couples. The third important issue in any co-localisation analysis is the topographical organisation of molecules within the cell that can frequently bias the interpretation of co-localisation coefficients. Indeed, confined molecules can strongly overlap or be very close (localisations) even if randomly distributed. To evaluate the statistical significance of the computed co-localisation index, most of the methods rely on pixel, spot or localisation randomisation inside the mask of the cell and compute empirical $p$-values by comparing the index measured experimentally to index obtained with simulations (Table 1). However, randomisation can be computationally expensive (especially for spot randomisation), and the robust interpretation of co-localisation indexes in terms of objects' (spots' or localisations') couples, i.e., significantly close to each other given the density of objects and the geometry of the cell, remains an open question.

We report the development of a method and software named Statistical Object Distance Analysis (SODA) to characterise the relative spatial positioning of several distributions of molecules, in a quantitative and automatic manner. Contrary to previous co-localisation methods, SODA allows to map statistically coupled objects within the cell, i.e., to compute a coupling probability for each single pair of objects. As SODA uses a marked-point process framework, it does not depend on the PSF characteristics and is robust to noise. In addition, by analysing the morphology (size, intensity and shape) and the distance separating coupled molecules, it provides, at a population level, a detailed and exhaustive description of the molecular assemblies. We validate the robustness and accuracy of SODA on simulated and synthetic fluorescence images where the coupling parameters are known, and we show that it outperforms standard correlation methods on well-studied biological examples.

Using three-colour structured-illumination microscopy (SIM) images of primary hippocampal neurons, we analyse the apposition of PSD95 and Homer, two abundant dendritic molecules, with Synapsin. After having integrated SODA in the open-source and freely available platform Icy[17], we automatically map more than 15,000 individual synapses. We observe an important diversity in the molecular composition and spatial arrangement of synaptic assemblies. By analysing the shape and the distances between molecules, we report that PSD95 and Homer lay in distinct postsynaptic functional nanodomains and are arranged in an asymmetric triangle with Synapsin.

We then image presynaptic glutamatergic terminals with three-dimensional stochastic optical reconstruction microscopy (STORM)[14] and analyse the coupling between more than 180,000 localisations of vesicular Glutamate Transporter (VGLUT) and Synapsin molecules inside putative synaptic boutons. We observe that each Synapsin or VGLUT localisation is in relation with ~5 copies of the other molecule, at a mean coupling distance of $52 \pm 18$ nm (s.d.). These two findings are in line with the known co-localisation of numerous copies of these two molecules on synaptic vesicles[18,19].

These results demonstrate that SODA is a versatile and effective tool to statistically map large data sets of multi-colour molecular assemblies in cellulo with high spatial resolution.

## Results

**SODA.** SODA can be used either on conventional microscopy images that contain clusters of molecules (wide-field, confocal, SIM, or STED) or with single-molecule-based microscopy (e.g., STORM, PALM or DNA-PAINT) where localisations of single molecules give direct access to their coordinates. To detect the fluorescent spots of aggregated molecules that are significantly brighter than the cell background and extract their coordinates on conventional microscopy images (Fig. 1a–d), we use a robust and automatic algorithm based on a wavelet transformation of the image and statistical thresholding of wavelet coefficients[9] (implemented as a plugin *Spot Detector* in the open-source image analysis software Icy (http://icy.bioimageanalysis.org/)[17]). These spots correspond to clusters of molecules where centre of mass (or intensity) is used as position. Note that in the case of localisation-based microscopy, the spatial coordinates are directly available. To statistically characterise the spatial distribution of molecules inside cells, we model the positions of single or aggregated molecules with a Marked Point Process[20], where the Mark is the ensemble of attributes of each individual fluorescent spot (colour, size, shape…, see Fig. 1d), and the Point Process is a mathematical model where the localisations of spots are viewed as a collection of points randomly located inside the cellular region of interest (ROI). Point processes are powerful statistical tools for modelling and analysing spatial data that have demonstrated their strength in such diverse disciplines as forestry[21], cell biology[22] or neurosciences[23]. To characterise the spatial relations between two populations $A_1$ (green) and $A_2$ (red) of objects (spots or localisations), we use the Ripley's K function[24], a gold standard for analysing the second-order properties (i.e., distance to neighbours) of point processes. For a distance parameter $r$, the function $K(r)$ is proportional to the number of $A_2$ objects (red typically) that are situated within a distance $r$ from $A_1$ (green) objects (Fig. 1d). K function also contains a boundary term that corrects for the possible under counting of neighbours near the boundary of the ROI[21] (see Supplementary Methods).

For a given distance parameter $r$, the $K(r)$ function counts all the pair of objects closer than $r$, and it is therefore difficult to precisely extract the distances where coupled objects accumulate. To count the number of (red) objects at specific distance from

**Table 1 Principal co-localisation index**

| Name | Type | Statistics | Pros | Cons | References |
|---|---|---|---|---|---|
| Pearson Correlation Coefficient (PCC) | Correlation | Pixel scrambling or Analytics | Easy to use; Works with any type of signal (diffuse, spotty, filaments...) | Depends on microscope resolution; Does not apply to localisation-based microscopy; Hardly interpretable in terms of spots' coupling | 4,5,11 |
| Cross-Correlation Spectroscopy | Correlation | Pixel scrambling or Analytics | Easily interpretable; Works with any type of signal | Depends on microscope resolution; Does not apply to localisation-based microscopy; Sensitive to local variations of intensity | 65,66 |
| DeBias | Correlation | Pixel scrambling | Allows to separate global bias from local interactions; Works with any type of signal | Depends on microscope resolution; Does not apply to localisation-based microscopy | 67 |
| Manders Overlap Coefficient (MOC) | Overlap | Spots' randomisation or Analytics (for ideal disk-shape spots[68]) | Easily interpretable; Works with any type of signal | Depends on microscope resolution; Does not apply to localisation-based microscopy; Randomisation can be computationally expensive | 3,6,12 |
| Thresholded Overlap (TO) | Overlap | Spots' randomisation or Analytics (for ideal disk-shape spots[69]) | Same as MOC, Possible selection of individual *coupled* spots with thresholded-overlap | Same as MOC, one more tunable parameter (threshold) | 8 |
| Mass-centre inside Mask (MM) | Distance-based (1st order, measures the density of (red) points in (green) masks) | Analytics | Same as TO | Depends on microscope resolution; Does not apply to localisation-based microscopy | 3 |
| Distance to Nearest-Neighbour (NN) | Distance-based (2nd order, measures the distance between neighbours) | localisations' randomisation | Does not depend on microscope resolution; Apply to localisation-based microscope | Global index (interaction strength[15] or False Discovery Rate[10]) hardly interpretable in terms of coupling; Apply to spotty objects only | 10,15 |
| Co-clustering of localisations | Correlation between localisations' clusters | localisations' randomisation | Apply to localisation-based microscopy | Hardly interpretable in terms of coupling; Not robust to mean coupling distance >0 | 55,56 |
| SODA | Distance-based (2nd order) | Analytics | Same as NN; Statistical mapping of individual couples of objects (spots, localisations) | Apply to spotty objects only | This study |

(green) objects, we introduce the function $\mathbf{G} = [K(r_{i+1}) - K(r_i)]_{i=0\ldots N-1}$, composed by incremental subtractions of the $K$ function for a series of increasing concentric distances $r_0 = 0 < r_1 < \cdots < r_N$ (Fig. 1d). $G$ is actually the (discrete) pair-correlation function with a boundary correction term, and is proportional to the number of $A_2$ (red) objects that fall inside the different $Ring(r_i, r_{i+1})$ around $A_1$ (green) objects (see Table 2 for the detailed definitions of variables and expression of $\mathbf{G}$).

The random distribution of (red) objects (null hypothesis) leads to stochastic fluctuations of their number inside each ring around (green) objects. Thus, an essential pre-requisite for the statistical quantification of the coupling between objects is the characterisation of the probability law of $\mathbf{G}$ under the null hypothesis of randomness. For a sufficiently large number of objects ($\geq$100 typically[5,16]), which is reached in most experiments, each component $G_i$ of vector $\mathbf{G}$ is normally distributed with mean $\mu_i$ and s.d. $\sigma_i$. Due to the linearity of the expected value, the mean of $G_i$ is equal to the mean of $K(r_{i+1})$ minus the mean of $K(r_i)$. Therefore, using the Ripley's boundary correction, $\mu_i$ is equal to the area (volume in three-dimensions) of the $Ring(r_i, r_{i+1})$ that is $\mu_i = \pi(r_{i+1}^2 - r_i^2)$ in 2D, and $\mu_i = \frac{4}{3}\pi(r_{i+1}^3 - r_i^3)$ in 3D[25]. The s.d. $\sigma_i$ depends on the covariance between $K(r_i)$ and $K(r_{i+1})$ and requires mathematically involved computations (see Supplementary Methods). When objects are statistically coupled, the number of (red) objects will be significantly enriched in a subset of rings around (green) objects. To statistically detect the rings where coupled objects accumulate and estimate the coupling probability of each individual pair of (green and red) objects, we

use the reduced Ripley's vector $\mathbf{G^0} = \frac{1}{\sigma}\mathbf{A}^{-1}.[\mathbf{G} - \boldsymbol{\mu}]$ with $\mathbf{A}$ a matrix that corrects for the possible overlap between rings around different (green) objects (Material and Methods). Under the null hypothesis of (red) objects' randomness, $\mathbf{G^0}$ is a Gaussian with zero mean and unit variance, and objects' coupling leads to a significant increase of a subset of $\mathbf{G^0}$ components, depending on the coupling distance. To determine the significantly positive components of $\mathbf{G^0}$, we use the procedure of Donoho and Johnstone[26] and hard-threshold the components $\mathbf{G^0}$ with the universal threshold $T(N) = \sqrt{2\log(N)}$, with $N$ the length of vector $\mathbf{G^0}$. Significant components of $\mathbf{G^0}$ that contain (red) couples are thus the components $G_i^0$ above the threshold $G_i^0 > T(N)$, and the coupling probability $P(\mathbf{x}, \mathbf{y})$ between a (green) object located at position $\mathbf{x}$ and a (red) object located at position $\mathbf{y}$ is equal to (see Material and Methods)

$$P(\mathbf{x}, \mathbf{y}) = \sum_{i=0}^{N-1} \mathbf{1}\{r_i < d(\mathbf{x}, \mathbf{y}) \leq r_{i+1}\} \frac{\sigma_i G_i^0 \mathbf{1}\{G_i^0 > T(N)\}}{G_i}. \quad (1)$$

where $\mathbf{1}\{.\}$ is the characteristic function that is equal to 1 when the condition (inequality) between the brackets is true, and is equal to 0 if not. (Red) objects with a coupling probability equal to 0 for any (green) object are single objects. On the other hand, objects with probability close to 1 are almost surely coupled. The total number of couples is equal to the overall sum of coupling probabilities $\sum_{\mathbf{x},\mathbf{y}} P(\mathbf{x}, \mathbf{y})$ and we define the global coupling index

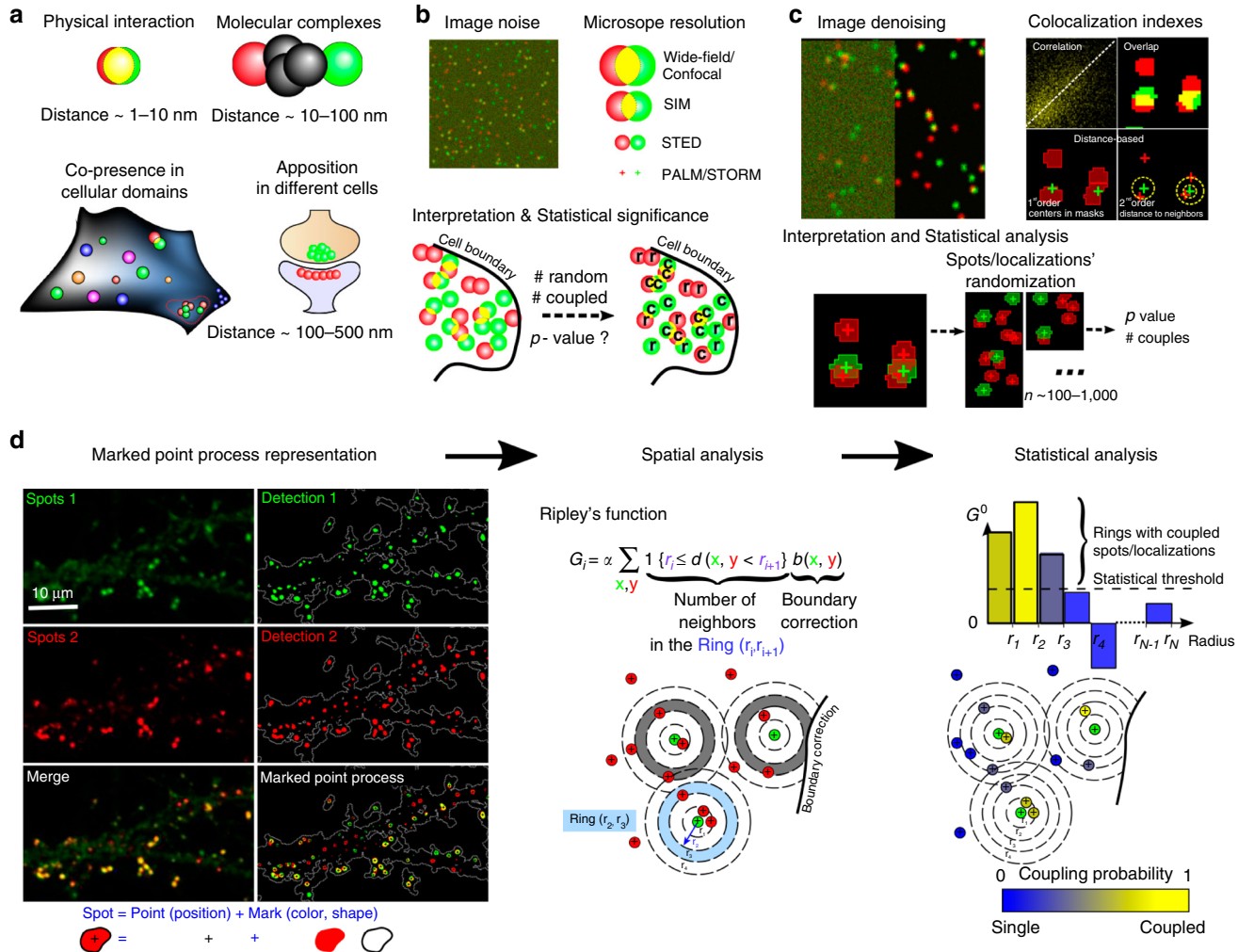

**Fig. 1** Co-localisation analysis of molecules' coupling. **a** Molecules' coupling embraces direct interaction (distance <10 nm), indirect interaction inside a macromolecular complex (distance between 10 and 100 nm), co-presence in cellular domains and synaptic apposition (distance between 100 and 500 nm). **b** Issues in co-localisation analysis are: (i) the sensitivity of co-localisation coefficients to noise, (ii) the dependence of methods on microscope resolution and (iii) the unbiased interpretation and statistical significance of co-localisation coefficients, as signal overlap/correlation can happen *by chance* for randomly distributed spots. Here for example, labels r and c designate random and coupled spots respectively. **c** Main steps of co-localisation analysis are: (i) image denoising and spots' extraction, (ii) quantification of the co-localisation of fluorescent signals, which can be evaluated with various techniques (correlations methods, physical overlap, distance-based index, see Table 1 for a detailed review of indexes), and (iii) statistical analysis of measured index with pixel/spots' randomisation. **d** SODA principles: (i) Molecules are labelled with different fluorescent probes (Homer (green spots 1) and PSD95 (red spots 2) are observed here with a confocal microscope). Fluorescent spots are automatically detected and represented with a Marked Point Process: the point is the spot's localisation (centre of mass or intensity) and the mark embraces morphological properties as the size and the colour of the spot. The ROI boundary is highlighted with a white dashed line. (ii) Spatial coupling between spots 1 and 2 is quantified with the Ripley's K function that counts the number of spots in channel 2 (red spots) that are in concentric rings around channel 1 spots (green spots) (Material and Methods, and Table 2). A boundary term corrects the under counting of neighbours near the boundary. (iii) Statistical thresholding (black dashed line) of the (reduced) Ripley's function indicates the rings where (red) spots 2 accumulate significantly. The number of coupled spots in each ring is proportional to the overshoot of the Ripley's function over the threshold. The coupling probability for each pair of spots is deduced from the ratio between the number of coupled (red) spots and the total number of (red) spots in each ring

of each population of objects as

$$\text{Coupling Index}(A_i) = \frac{1}{n_i} \sum_{\mathbf{x},\mathbf{y}} P(\mathbf{x},\mathbf{y}), \qquad (2)$$

with $n_{i=1,2}$ the total number of objects within population (Point Process) $A_1$ or $A_2$. Using the coupling probabilities between all the individual pairs of objects, we can statistically analyse the morphological parameters of objects with respect to their coupling properties, and measure for example size or shape of coupled objects compared to single ones. Moreover the mean

distance between coupled objects is given by the probability-weighted sum

$$\text{Mean Coupling Distance} = \frac{\sum_{\mathbf{x},\mathbf{y}} [P(\mathbf{x},\mathbf{y}) d(\mathbf{x},\mathbf{y})]}{\sum_{\mathbf{x},\mathbf{y}} P(\mathbf{x},\mathbf{y})}. \qquad (3)$$

**Validation of SODA**. In a first step, we use synthetic fluorescence images to test the accuracy and statistical robustness of SODA (Fig. 2a). Images are generated using a mixed Poisson-Gaussian model (Material and Methods) and signal-to-noise ratio (SNR) is

**Table 2 Mathematical variables**

| Name | Mathematical Expression | Meaning |
|---|---|---|
| Point-process $i = 1, 2$ | $A_{i=1,2}$ | Positions of all the objects (spots or localisations) $i = 1, 2$ |
| Number of objects $i = 1, 2$ | $n_{i=1,2}$ | Number of objects in $A_{i=1,2}$ |
| Distance between objects | $d(\mathbf{x}, \mathbf{y})$ | Distance between (green) object located at position $\mathbf{x}$ and (red) object located at $\mathbf{y}$ |
| Boundary correction | $k(\mathbf{x}, \mathbf{y})$ | Corrects the under-estimation of object's neighbors near the ROI boundary (Supp. Methods) |
| Ripley's K function | $K(r) = \frac{Volume\{ROI\}}{n_1 n_2} \sum_{x,y} 1_{\{d(x,y) \leq r\}} k(x, y)$ | Counts the number of (red) objects at a distance below $r$ from (green) objects |
| Searching distances | $0 = r_0 < r_1 < \cdots < r_N$ | Increasing distances around (green) objects where the K function is computed |
| Rings | $Ring(r_i, r_{i+1})$ | Sub-region of the ROI that contains points ($\mathbf{y}$) located at a distance $r_i \leq d(\mathbf{x}, \mathbf{y}) \leq r_{i+1}$ from a (green) object ($\mathbf{x}$) |
| Ripley-based vector | $G = [K(r_{i+1}) - K(r_i)]_{0 \leq i \leq N-1}$ | Counts the number of (red) objects inside concentric rings around (green) objects |
| Number of rings | $N$ | Number of rings and length of the vector G |
| Mean of **G** | $\mu = [\mu_i]_{0 \leq i \leq N-1}$ with $\mu_i = \pi(r_{i+1}^2 - r_i^2)$ (2D) or $\mu_i = \frac{4}{3}\pi(r_{i+1}^3 - r_i^3)$ (3D) | Expected mean of G under the null hypothesis of $A_2$ randomness |
| Standard deviation of **G** | $\sigma = [\sigma_i]_{0 \leq i \leq N-1}$ | Standard deviation of G under the null hypothesis of $A_2$ randomness (see Supplementary Methods) |
| Rings' overlapping matrix | $\mathbf{A} = [\alpha_{i,j}]_{0 \leq i,j \leq N-1}$ with, $\alpha_{i,j} = \frac{Volume\{Ring(r_i, r_{i+1}) \cap Ring(r_j, r_{j+1})\}}{Volume\{Ring(r_i, r_{i+1})\}}$ | Proportion of the volume of $Ring(r_i, r_{i+1})$ that overlaps with $Ring(r_j, r_{j+1})$ |
| Reduced Ripley-based vector | $G^0 = \frac{1}{\sigma}A^{-1}.[G - \mu]$ | Reduced Ripley-based vector with zero mean and unit variance (under the null hypothesis of $A_2$ randomness) |
| Statistical threshold | $T(N) = \sqrt{2 \log(N)}$ | Statistical threshold to extract rings with coupled (red) objects. |
| Number of couples per ring | $C = \left[1_{G_i^0 \geq T(N)} \frac{n_1 n_2}{Volume\{ROI\}}(G_i - \mu_i)\right]_{0 \leq i \leq N-1}$ | Statistical estimate of the number of couples per ring. |
| Couples without overlapping | $\widetilde{C} = A^{-1}.C = \left[1_{G_i^0 \geq T(N)} \frac{n_1 n_2 \sigma_i}{Volume\{ROI\}} G_i^0\right]_{0 \leq i \leq N-1}$ | Number of couples corrected for rings' overlapping. |
| Number of pairs | $\frac{n_1 n_2}{Volume\{ROI\}} G$ | Total number of object pairs inside rings. |
| Coupling probability | $P(x, y) = \sum_{i=0}^{N-1} 1_{r_i \leq d(x,y) < r_{i+1}} \frac{1_{G_i^0 \geq T(N)} \sigma_i G_i^0}{G_i}$ | Probability that a (green) object located at position x is coupled with a (red) object located at y |
| Coupling index | $Coupling\ Index(A_i) = \frac{1}{n_i} \sum_{x,y} P(x, y)$ | Mean number of coupled objects (i.e., probability-weighted) in each population $A_{i=1,2}$ |
| Mean coupling distance | $Mean\ Coupling\ Distance = \frac{\sum_{x,y}[P(x,y)d(x,y)]}{\sum_{x,y} P(x,y)}$ | Probability-weighted distance between coupled objects |

either set to SNR = 3 (low) or SNR = 6 (high). The distances between coupling spots follow a Gaussian process (Material and Methods) with fixed s.d. $\sigma = 0.3$ pixels and increasing mean 0, 1 and 2 pixels. We compare SODA performances with those of most of the existing co-localisation methods (Table 1): Pearson Correlation Coefficient (PCC), Manders Overlap Coefficient (MOC), Thresholded Overlap (TO) and (first-order) distance-based Mass-centre within Mask (MM) index. We did not compare SODA to other known distance-based methods such as Gibbs[15] and false positive rate (FPR)[10] index because their index cannot be easily related to the number of couples and compared with the other methods. We observe that SODA is accurate and is the only method that is robust to changes in SNR and coupling distance. Indeed, and for the same spots' detection and segmentation, correlation (PCC), overlap (MOC and TO) and distance-based (MM) index decrease with SNR and when the distance between coupled spots increases. This is mainly due to decreased overlap between segmented spots and, for the distance-based MM method, because the probability that the centre of mass of (red) masks lay inside (green) masks decreases with the coupling distance. We next measure the statistical power of SODA and the accuracy of its estimation of the coupling distance (Fig. 2b). We observe that SODA $p$-value (Material and Methods) decreases rapidly with the simulated percentage of coupling and drops below 1% for a coupling percentage above 5% for every SNR and distance. On the other hand, the accuracy of coupling distance estimation increases both with SNR and simulated distances. This

is due to the decreased precision of spots' localisation for low SNR (Supplementary Fig. 1). Moreover, because the localisation error remains upper-bounded by ≈0.5 pixel for SNR ≥ 3, SODA's estimates are more accurate for high coupling distance (above 1 pixel), as the localisation error becomes negligible.

To measure the performances of SODA irrespective of the precision of spots' detection and localisation, we then perform point process simulations where spots are replaced by localisations (Fig. 2c). We observe that SODA's measures of coupling match perfectly the simulated coupling values at any coupling distance. Moreover, the estimates of coupling distances are also much more accurate and for complete localisations' coupling, estimated distances approach the ground truth. These results with point processes show that discrepancies between SODA estimates and ground truth in synthetic fluorescence images are mainly coming from mis-detection and mis-localisation of fluorescent spots rather than spatial analysis, and indicate that SODA is a very robust methods for analysing localisation-based microscope images. Finally, using point process simulations, we make several parameters (number of localisations, coupling distance…) vary and further explore the performances of SODA (Supplementary Fig. 2). We demonstrate that SODA remains very robust and accurate for a wide range of parameters, even for high coupling distances and high localisation densities.

Finally, we validate SODA robustness on known biological examples. For this, we use total internal reflection fluorescence (TIRF) microscopy and analyse the coupling at the cell

membrane between two well-characterised cargos, transferrin (Tf) and interleukin 2 receptors (IL-2R), with proteins implicated in different endocytic pathways (Fig. 2d). Tf and IL-2R are internalised into distinct domains of the plasma membrane: Tf is taken up exclusively into clathrin-coated pits[27], whereas IL-2R are clustered into cholesterol enriched microdomains, devoid of clathrin[28]. As expected in the negative control, SODA does not detect coupling ($p$-value = 0.085, Material and Methods) between clathrin and IL-2R. However, and with the same segmented spots, Manders ($p$-value = $1.2{\times}10^{-3}$, pixel scrambling) and Pearson ($p$-value = $2.8{\times}10^{-6}$, pixel scrambling) coefficients overestimate the coupling between clathrin and IL-2R fluorescent signals. These false positive detections of coupling are likely due to the fortuitous overlap between spot masks, and the correlation between background intensities[5]. In the positive control, the three methods measure an important, comparable and statistically relevant coupling ($p$-value $\leq 10^{-16}$) between Tf and clathrin-coated structures. Using SODA, we also measure a mean coupling distance of $1.53 \pm 0.09$ pixels (s.d.) (i.e., $91.8 \pm 5.4$ nm (s.d.)) between Tf and clathrin spots. As the SNR is approximately equal to 6, the localisation error can be neglected compared to the measured coupling distance (Supplementary Fig. 1). The computed coupling distance <100 nm indicates that Tf lay inside clathrin-coated structures with a reported diameter of approximately 200–500 nm[29]. Altogether, these results demonstrate that SODA is robust in negative and positive biological controls, and allow to unravel important features such as distances between coupled spots.

**Mapping the glutamatergic synapses with SIM.** Synapses are specialised contact sites where finely regulated molecular interactions mediate neuronal communication. Due to different neurotransmitters and functions (inhibitory/excitatory), and constant remodelling (plasticity)[30], synapses present a high diversity in morphology and composition. Thus, the robust analysis of these molecular assemblies is essential to unravel their function and determine how a single dendrite integrates the inputs coming from hundreds to thousands of neurons[31]. Although many studies focused on the analysis of the spatial organisation and the molecular composition of synapses[32–35], they were limited to a few dozens of synapses in the best case because more powerful methods were missing.

Synapses are composed of the presynaptic bouton which contains the synaptic vesicles, and the postsynaptic compartment located on the target neurons where receptors and anchoring proteins accumulate and form the postsynaptic density. Neuronal proteins are synthesised in the cell body, transported in long tube-shaped axons and dendrites before eventually accumulating at the synapse. Dendritic proteins can thus be found as immunofluorescent spots in front of presynaptic terminals (postsynaptic localisation) or at extrasynaptic sites during transport or development[36]. Here, we apply SODA to triple-labelled structured-illumination microscopy (SIM) images and map the molecular arrangement of three major molecules constituting the glutamatergic synapses: the Synapsin that tethers the reserve pool of presynaptic vesicles to actin[37], the post-synaptic molecule PSD-95 that anchors *N*-methyl-D-aspartate (NMDA)[38] and stabilises $\alpha$-amino-3-hydroxy-5-methyl-4-isoxazolepropionic acid (AMPA)[39] ionotropic glutamate receptors, and Homer that anchors metabotropic glutamate receptors[40] (Fig. 3a).

Based on the localisation of molecules (intensity centres of detected spots), we compute the coupling probability for each individual pair of presynaptic and postsynaptic spots (Eq. 1). To analyse automatically large data sets, we build a new batch

analysis that uses graphical programming in Icy (plugin Protocols, Fig. 3). This protocol allows us to automatically segment the neuronal shape, detect and localise spots, apply SODA and export results on a batch of SIM images (Material and Methods). For each picture, we thus identify single (isolated) and coupled spots, and construct the corresponding colour-map of molecule populations. For each spot or assembly, we also extract multiple morphological (size, shape and intensity) and spatial (localisation and coupling distances) information.

Among all the detected spots, we find that ~30% of dendritic spots and half of presynaptic Synapsin spots are single (Fig. 4a). We highlight that many single spots are small and were previously unobservable with wide-field or confocal microscopy. The larger proportion of single Synapsin spots is likely due to the apposition of a subset of Synapsin spots in front of unlabelled, inhibitory synapses[31,36,41]. We also measure that 25% of PSD95 spots and 40% of Homer spots are statistically apposed to Synapsin without the other dendritic molecule. Finally, we find that the majority of dendritic clusters are together apposed to Synapsin forming a ménage à trois. As PSD95 and Homer anchor different classes of glutamatergic receptors, all these synaptic assemblies should present different functional properties, and we thus investigate their morphological characteristics.

Single dendritic spots (without labelled partner) are mostly small and faint spots with a mean size similar to the SIM PSF = $0.010\ \mu m^2$. Isolated dendritic spots should thus correspond to small clusters in transport, as observed in fluorescence[42] and electron microscopy[43], but are rarely identified in confocal microscopy. Then, we observe that synaptic PSD95 and Homer are slightly bigger, but far smaller than extrasynaptic PSD95-Homer couples. Thus, synaptic localisation of the dendritic spots cannot be inferred from cluster size as suggested previously[44]. Actually, most of the 10% biggest dendritic clusters are indeed synaptic, and <0.3% of dendritic spots with a size > $0.05\ \mu m^2$ are isolated, and we show here that many small and faint dendritic spots are also apposed to synapsin. Within all PSD95-Homer couples, the sizes of PSD95 and Homer clusters are highly correlated (Pearson coefficient $R = 0.92$) even at extrasynaptic sites ($R = 0.78$) suggesting that dendritic molecular assembly of anchoring proteins is correlated independently of presynaptic inputs. Finally, in triplets, the size of postsynaptic clusters is even bigger and correlated ($R = 0.92$). Contrary to postsynaptic clusters, the size of Synapsin spots is nearly uniform indicating a constant number of tethered synaptic vesicles in the different synapses. We suggest that triplets could correspond to mature, potentiated synapses. Indeed, large PSD95 spots[45], together with high copy numbers of NMDA[46] and AMPA receptors[47], are associated with excitatory synapse maturation.

To map the geometry of synaptic assemblies, we measure distances between coupled spots. We find that postsynaptic anchoring molecules are much closer than apposed clusters. Distances between Synapsin and postsynaptic spots are similar to distances reported with localisation-based microscopy (mean axial distance between Bassoon, a major scaffold protein of the presynaptic active zone, and Homer = 153 nm[32]). However, we observe a high variability in coupling distances over the 15,000 synaptic assemblies, s.d. being twice as big as those reported with localisation-based microscopy on 127 synapses[32]. This increase exemplifies the larger variability at a population level compared to a small subset of synapses. Based on the triangular arrangement of the ménage à trois, we compute that the mean axial distance between PSD95 and Homer along the principal synaptic axis PSD95-Synapsin is equal to 15 nm, in accordance with the distance measured in electron microscopy = 22 nm[43]. Moreover the analysis of >7800 ménage à trois

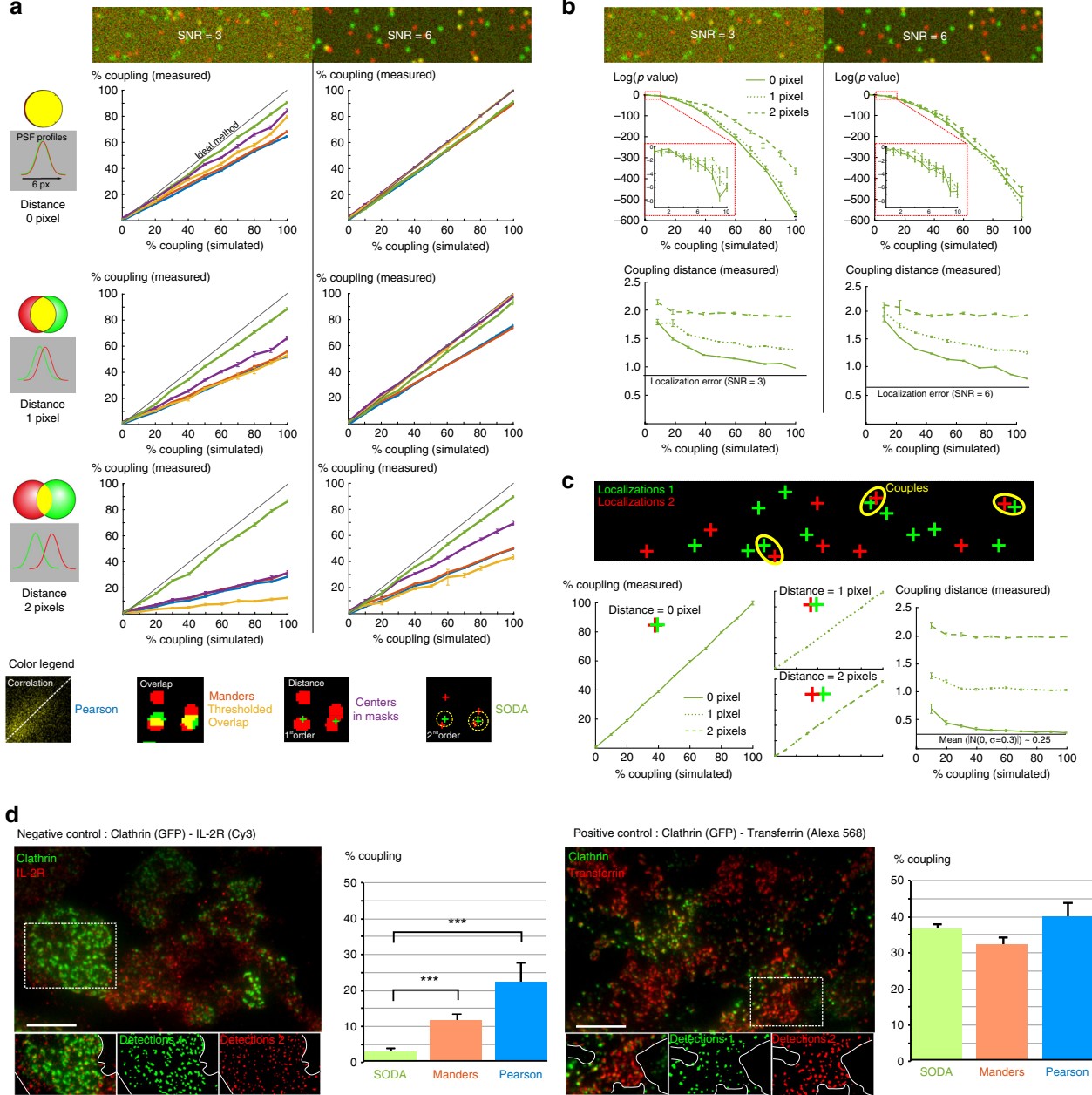

**Fig. 2** Validation of SODA **a** Synthetic fluorescent images with different SNR and coupling parameters are generated (Material and Methods). SODA is compared with main coupling indexes (see Table 1): Pearson correlation coefficient (PCC), Manders Overlap Coefficient (MOC), Thresholded Overlap (TO) with $T = 0.5$ (i.e., percentage of (red) spots whose more than 50% of the mask overlaps with a (green) mask[8]) and Mass-centre in Masks (MM) (error bars = ±1 standard error of the mean (s.e.m.), 10 synthetic images per condition). **b** (Log) $p$-value (Material and Methods) and mean coupling distance (Eq. 3) are computed with SODA for increasing coupling distances ($d = 0$ pixels in solid line, $d = 1$ in dotted line and $d = 2$ in dashed line). (Log) $p$-values for coupling percentages between 0 and 10% are zoomed (red dashed box). Error bars = ±1 s.e.m. **c** Testing SODA with point process (localisations) Monte-Carlo simulations (10 simulations per distance and coupling index). $n_1 = 100$ (red) points (=localisations) and $n_2 = n_1 = 100$ (green) points are distributed in a 256 × 256 square (Material and Methods). The expected mean distance ≈0.25 for a Gaussian point process with mean 0 and s.d. 0.3 (Material and Methods) is highlighted with a continuous black line. Error bars = ±1 s.e.m. **d** Analysis of the coupling between two endocytic cargos (IL-2R or Tf) and Clathrin (Hep2beta Clathrin-GFP). IL-2R, Tf (red) and Clathrin (green) molecules are labelled with fluorescent probes and observed in total internal reflection fluorescence (TIRF) microscopy. Fluorescent spots are automatically extracted using Spot detector in Icy. Cell boundaries are highlighted with a white solid line. The coupling index between IL-2R and Clathrin (negative control) estimated with SODA is compared with Pearson (PCC) and Manders (MOC) coefficients. Note that SODA does not detect coupling (percentage = 2.41 ± 0.6% (s.e.m.) ($p$-value = 0.085) between clathrin (13 cells, 5124 spots) and IL-2R (6145 spots), contrary to Manders (12.6 ± 1.04%, $p$-value with pixel scrambling = 0.0012) and Pearson correlation analysis (21.9 ± 5.97%, $p$-value with pixel scrambling = 2.8 $10^{-6}$). In the positive control, the three co-localisation index measure an important, comparable and statistically relevant coupling between Tf (15 cells, 8407 spots) and clathrin-coated structures (9623 spots) (SODA: 36.5 ± 1.49%, $p$-value = 1.54 $10^{-16}$; Manders: 31.7 ± 2.38%, $p$-value < $10^{-16}$ and Pearson: 40.8 ± 3.03%, $p$-value < $10^{-16}$). Error bars = 95% c.i. Scale bar = 10 μm

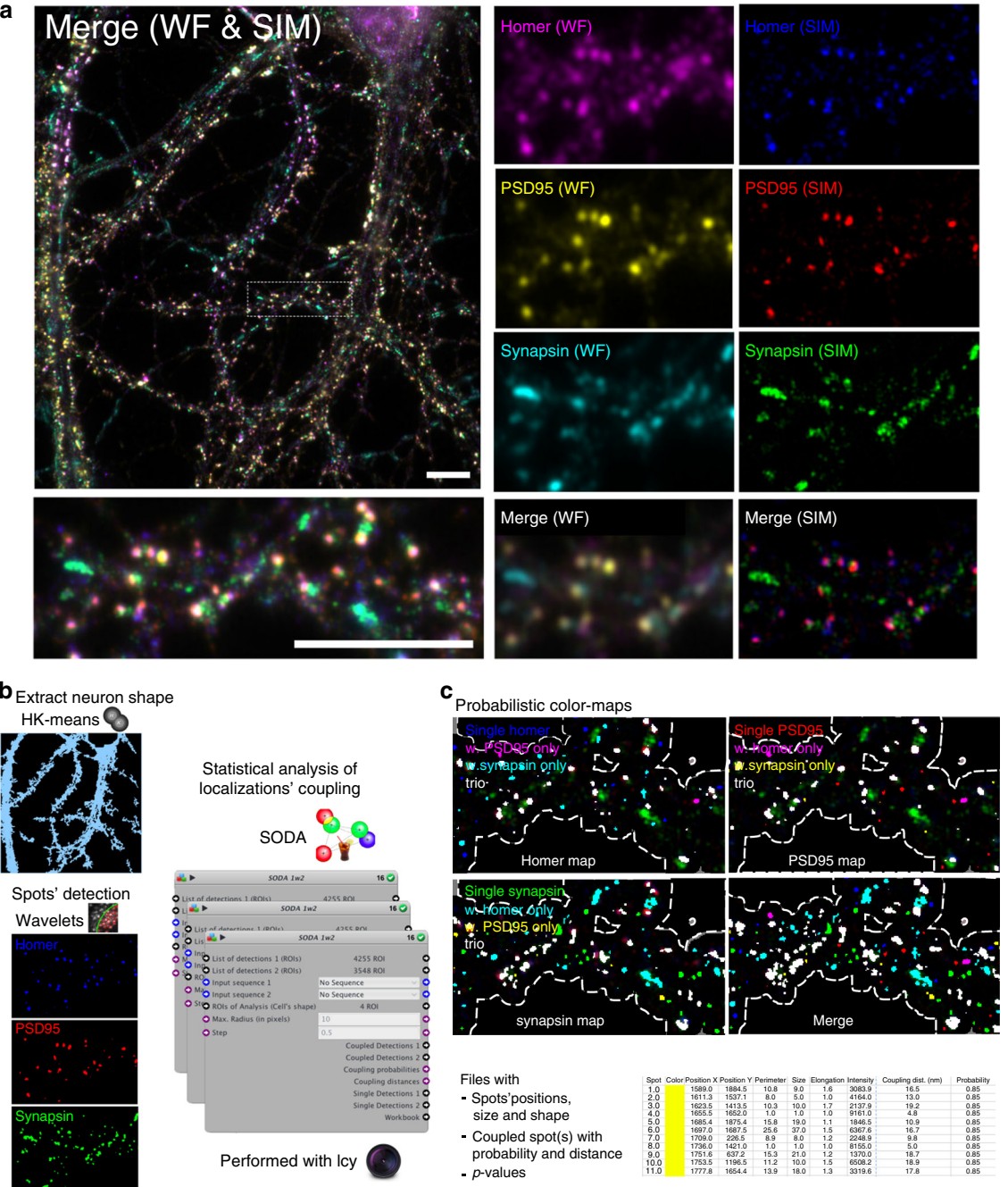

**Fig. 3** Batch analysis using graphical programming in Icy. **a** The input of SODA protocol is a folder that contains multiple three-colour SIM images of primary hippocampal neurons. Postsynaptic anchoring molecules PSD95 (red) and Homer (blue), and the presynaptic molecule Synapsin (green) are labelled. **b** The publicly available protocol consists of multiple elementary blocks that sequentially perform multiple image analysis. (i) Cell body and neuronal shape are isolated with two HK-means thresholdings of Homer labelling. Dendritic mask is then obtained by substracting the cell body mask to the neuronal mask. (ii) Spot detector blocks extract pre- and postsynaptic spots inside the dendritic mask. (iii) SODA blocks analyse the coupling between the localisations of PSD95, Homer and Synapsin spots statistically. The complete screenshot of the protocol is shown in Supplementary Fig. 3. **c** Outputs of SODA protocol are: (i) the probabilistic colour maps of PSD95, Homer and Synapsin spots, where different colour masks are associated to isolated (single) spots, coupled spots and triplets, and (ii) files where the positions of each spots, their individual morphology (size, intensity, shape...), the distance to eventual associated spot(s) and corresponding coupling probabilities are exported automatically. Scale bar = 10 μm

assemblies shows that Homer clusters are slightly peripheral compared to the PSD95-Synapsin principal axis, indicating an arrangement of the triplet PSD95-Homer-Synapsin in an asymmetric triangle.

To rule out any potential bias of our method in complex-shaped dendrites, we perform realistic point process simulations in the dendritic masks extracted from immunofluorescence images

(Fig. 4b). For this, we characterise the coupling distances between PSD95, Homer and Synapsin probabilistically, and find that histograms of coupling distances can be approximated accurately by a Gaussian point process (Material and Methods). Using the coupling parameters measured experimentally, we then simulate realistic synaptic coupling in the extracted dendritic masks. We compare SODA with the classical approach consisting of

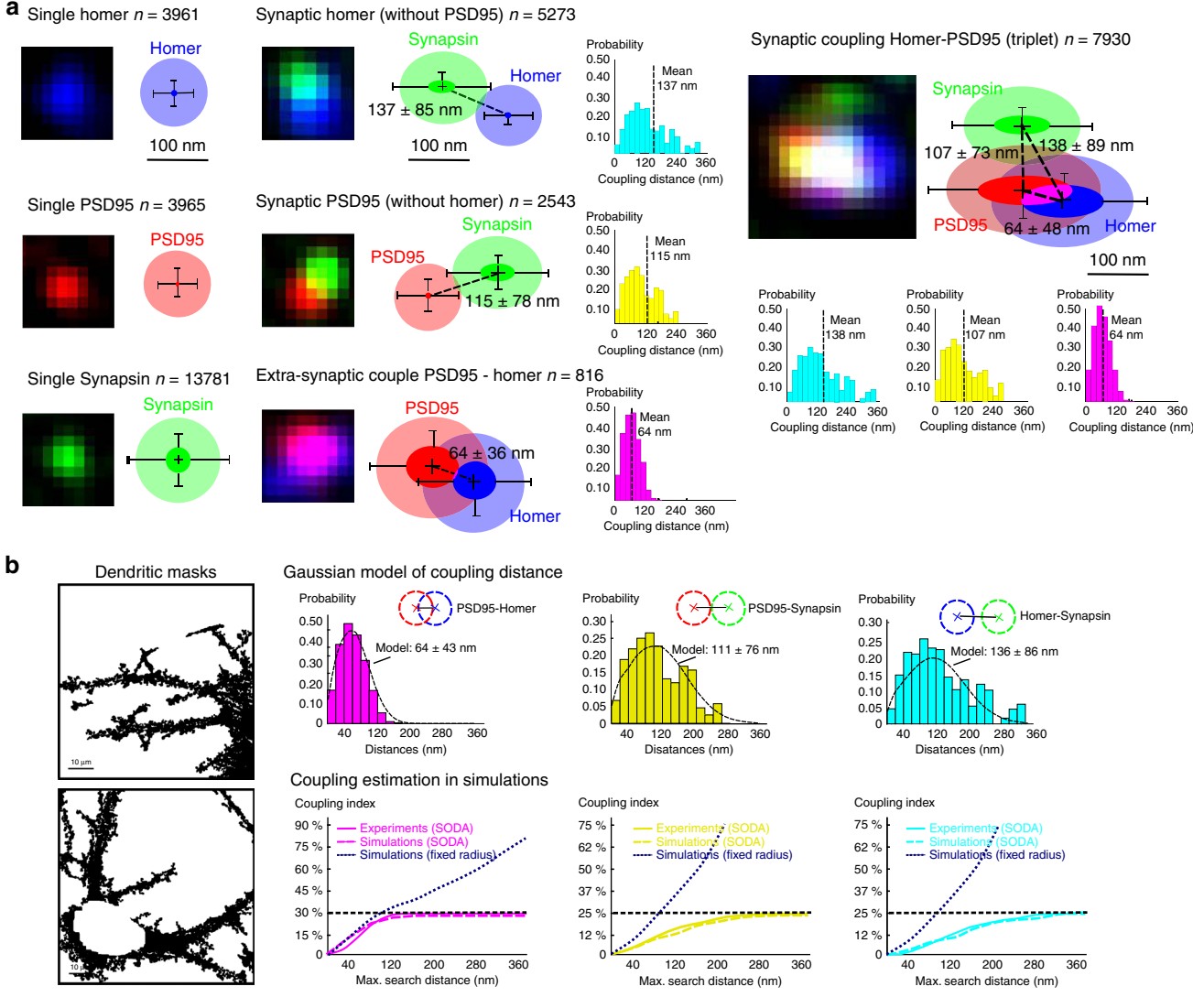

**Fig. 4** A statistical view on glutamatergic synapse morphometry. **a** Using SODA protocol, the coupling of $n = 11200$ PSD95, $n = 13359$ Homer and $n = 26505$ Synapsin spots among $N = 9$ neurons is mapped automatically. Single Homer ($n = 3961$), PSD95 ($n = 3965$) and Synapsin spots ($n = 13,781$) are extracted statistically. The average elliptic fit of spots is represented at scale (error bars = ±s.d.). The transparent part represents the PSF halo. For each molecular assembly, a representative fluorescent patch is shown. We also extract extrasynaptic couples of Homer-PSD95 ($n = 816$, coupling $p$-value $= 10^{-10}$, mean cluster size $= 0.025$–$0.028$ μm$^2$, s.e.m. $= 5.3 \times 10^{-5}$), and synaptic appositions of PSD95 ($n = 2543$, coupling $p$-value $= 10^{-40}$) and Homer spots ($n = 5273$, $p$-value $= 10^{-21}$) (Cluster size of synaptic PSD95/Homer $= 0.012$–$0.018$ μm$^2$, s.e.m $= 3.5 \times 10^{-5}$). For these three couples, we represent at scale the average morphology of molecular assemblies (error bars = ±s.d.), and the weighted histograms of coupling distances. Most synaptic assemblies ($n = 7930$) are composed by PSD95 and Homer apposed to Synapsin, forming a molecular triplet. In triplets, the size of dendritic clusters is even bigger (PSD95 mean size $= 0.043$ μm$^2$ and Homer mean size $= 0.038$ μm$^2$). Synapsin cluster size is smaller (mean ± s.d. $= 0.018$–$0.021$ μm$^2$, s.e.m. $= 9.3 \times 10^{-6}$) and similar to to the size of isolated Synapsin clusters and those solely apposed to either PSD95 or Homer. We represent the average morphology and spatial organization of ménage à trois assemblies (error bars = ±s.d.) and we plot the weighted histograms of coupling distances. Scale bar = 100 nm. **b** Validating the robustness of SODA with simulations. (i) Simulations inside the extracted dendritic masks are performed. Number of points (1500 PSD95 and Homer positions and 3000 Synapsin positions per dendritic mask) are similar to the observed objects (spots)' density. (ii) Coupling distances between PSD95, Homer and Synapsin spots are modelled with a Gaussian point process (Material and Methods). Simulated distances and coupling indexes (Homer-PSD95 = 35.4%, PSD95-Synapsin = 30.3% and Homer-Synapsin = 26.9%) are those measured experimentally with SODA. (iii) For increased searching distance, the SODA coupling index (dashed line) is compared with the measured index (solid line) and the index obtained by counting all the pairs of localisations within the search distance (dotted navy blue line)

counting all the pairs of spots closer than a pre-defined distance. We find that SODA reaches a plateau at search distances above 300 nm, contrary to the index obtained with the naive counting method that increases continuously due to the presence of false positive, random spots. We highlight that the plateau value of 300 nm is slightly higher than the traditional cut-off value (250

nm) to determine synaptic apposition[32], and that at this distance, synaptic apposition with classical approach is overestimated by more than 300%. As SODA estimates in experiments and simulations are very close, these findings indicate that SODA is robust even in complex-shaped dendrites with a high density of spots.

**Analysing coupling between single synaptic localisations.** SODA uses the framework of point processes to analyse spatial relations between objects. It is thus particularly well adapted to localisation-based microscopy where spatial coordinates of molecules are directly computed by the imaging software. To supplement our study, we image two presynaptic molecules with the 3D-STORM Vutara system (Fig. 5a and Supplementary Movie 1) (Material and Methods): the vesicular glutamate transporter (VGLUT) that is responsible for the uptake of the excitatory amino acid, L-glutamate, into synaptic vesicles[48], and Synapsin that is involved in the binding of synaptic vesicles to the cytoskeleton[49]. We then delineate automatically putative synaptic boutons and statistically map single and coupled individual localisations inside boutons.

The automatic delineation of biological structures using molecules' localisations is an important and challenging issue when using localisation-based microscopy. In most cases, the localisations are densely packed (clustered) into the structures of interest, while background intensity leads to isolated localisations that have to be screened out from the analysis. Most of the algorithms proposed so far to automatically segment domains with clustered localisations are based on Gaussian blurring[50], Voronoi tesselation[51], density-based (DBSCAN)[52] and Ripley-based clustering[53,54]. Here to robustly and rapidly segment the putative synaptic boutons where VGLUT and Synapsin localisations are clustered, we adapt the DBSCAN method and implement it as an Icy block inside our complete STORM protocol (Material and Methods and Supplementary Fig. 3). Thanks to this programme, we automatically process a batch of STORM images and statistically analyse nearly 500,000 localisations. We find that ~80% of VGLUT localisations (=76.9% of 244,410 localisations) and Synapsin localisations (83.0% of 171,719 of localisations) are inside the automatically segmented clusters. We then pick putative synaptic boutons as the boolean intersection of VGLUT and Synapsin clusters. More than half of the clustered VGLUT (52.7%) and Synapsin (60.0%) are inside boutons (defined as clusters' intersections) indicating that molecules localisations lay in highly overlapping presynaptic volumes.

The very high density of molecule localisations makes any guess or manual picking of coupled localisations nearly impossible. Different co-localisation methods have been proposed in localisation-based, super-resolution microscopy[55,56]. These methods delineate localisation clusters with second-order spatial analysis (nearest-neighbour or Ripley K function), before measuring the overlap between the clusters in different colours with standard correlation coefficients (Spearman or Pearson). While these methods have the advantage that they can be applied to single localisations, the empirical computation of the statistical significance of each correlation coefficient would need multiple randomisations of the tens of thousands single localisations in each image. Moreover, these methods do not measure the coupling properties (probability and distance) between individual pairs of localisations (Table 1). We thus integrated a SODA block in our protocol that computes automatically the individual coupling probabilities between all the pair of localisations inside putative synaptic boutons, and maps the positions of single and coupled localisations (Fig. 5b and Supplementary Movie 1). We find that 57.8% of the VGLUT localisations inside putative boutons are coupled with 62.5% of the Synapsin localisations, and that the mean coupling distance between all the localisations couples is equal to 52 nm. We also observe that the coupling distance does not exceed 80 nm. The overall coupling index is equal to 11.2%, which is highly significant ($\log_{10}(p\text{-value}) = -57$, Material and Methods), though individual coupling probabilities are quite low

$(3.4 \pm 1.9\%$ (s.d.)). To reach a significant coupling index, low individual probabilities are counterbalanced by a high coupling stoichiometry. Indeed, we find that each coupled VGLUT localisation is associated with a mean of 4.69 Synapsin localisations, and that each coupled Synapsin is associated on average with 4.96 VGLUT localisations. Coupling stoichiometry is thus approximately equal to 5:5.

The previous results are in line with the co-presence of numerous copies of VGLUT and Synapsin localisations around tiny synaptic vesicles with outer diameter ~40 nm[19]. Moreover, the measured coupling stoichiometry is in accordance with the average 8 Synapsin and 9 VGLUT1 molecules per synaptic vesicles that has been previously reported with purification and mass-spectrometry analysis[19,57]. The slightly lower stoichiometry that we report might be due either to unlabelled molecules, or to the fact that previous proteomic analysis were either performed on entire brains[19] or on the cerebellum and cortex of adult rat brains[57], while we focus here on the molecular organization of rodent hippocampi. It should be noted that stoichiometry estimated with SODA is directly dependent on the STORM labelling efficiency (antibody affinity, blinking efficiency, number of acquired images…). Thus, one should be careful and use high-affinity and well-characterised primary and/or secondary antibodies to get a robust estimation of stoichiometry.

Altogether these results demonstrate the capability of SODA to analyse robustly several thousands of densely packed localisations, statistically map their coupling and describe nanometre scale assemblies such as synaptic vesicles in cellulo.

To test and prevent any bias of SODA, we perform point process simulations inside the putative boutons delineated experimentally (Fig. 5c). We first model the coupling distance between VGLUT and Synapsin localisations with a Gaussian process (Material and Methods). We then simulate Synapsin localisations with known coupling index and distances around experimental VGLUT localisations inside delineated boutons. First, we observe that SODA has a good statistical specificity, even when localisations are densely packed, as it does not detect any coupling when Synapsin localisations are randomly distributed inside putative boutons ($p\text{-value} = 0.30 \pm 0.09$ (s.d.), Material and Methods). On the other hand, SODA is also very sensitive as $p$-values rapidly drop when the simulated coupling index increases. Moreover, we observe that SODA is also accurate as it measures a coupling index close to the simulated ground truth. The slight underestimation of the coupling index is likely due to the very high density of localisations, combined with an important variability (s.d.) of the coupling distance (Supplementary Fig. 2a, e).

Overall, these simulation results demonstrate the accuracy and statistical robustness of SODA in measuring the coupling between densely packed localisations in 3D STORM images.

## Discussion

SODA uses the localisation of molecule spots in high-resolution fluorescence microscopy and object-based statistics to map the diversity of molecular assemblies at a population level. Because we compute explicitly the statistical properties of the Ripley's function, SODA does not require any computer simulation to test the randomness of spot distributions and compute an unbiased coupling index between localisations. This diminishes drastically the computation time, and while the rate-limiting step of the algorithm is the computation of the Ripley's K function at different distances it just requires a number of elementary operations that scales linearly with the number of spots. We also optimise the computation of Ripley's K function with image partitioning, so that it only takes a running time of 30–50 s on a single core i7 (2.0 GHz) to analyse a 1900 × 1900 pixels SIM

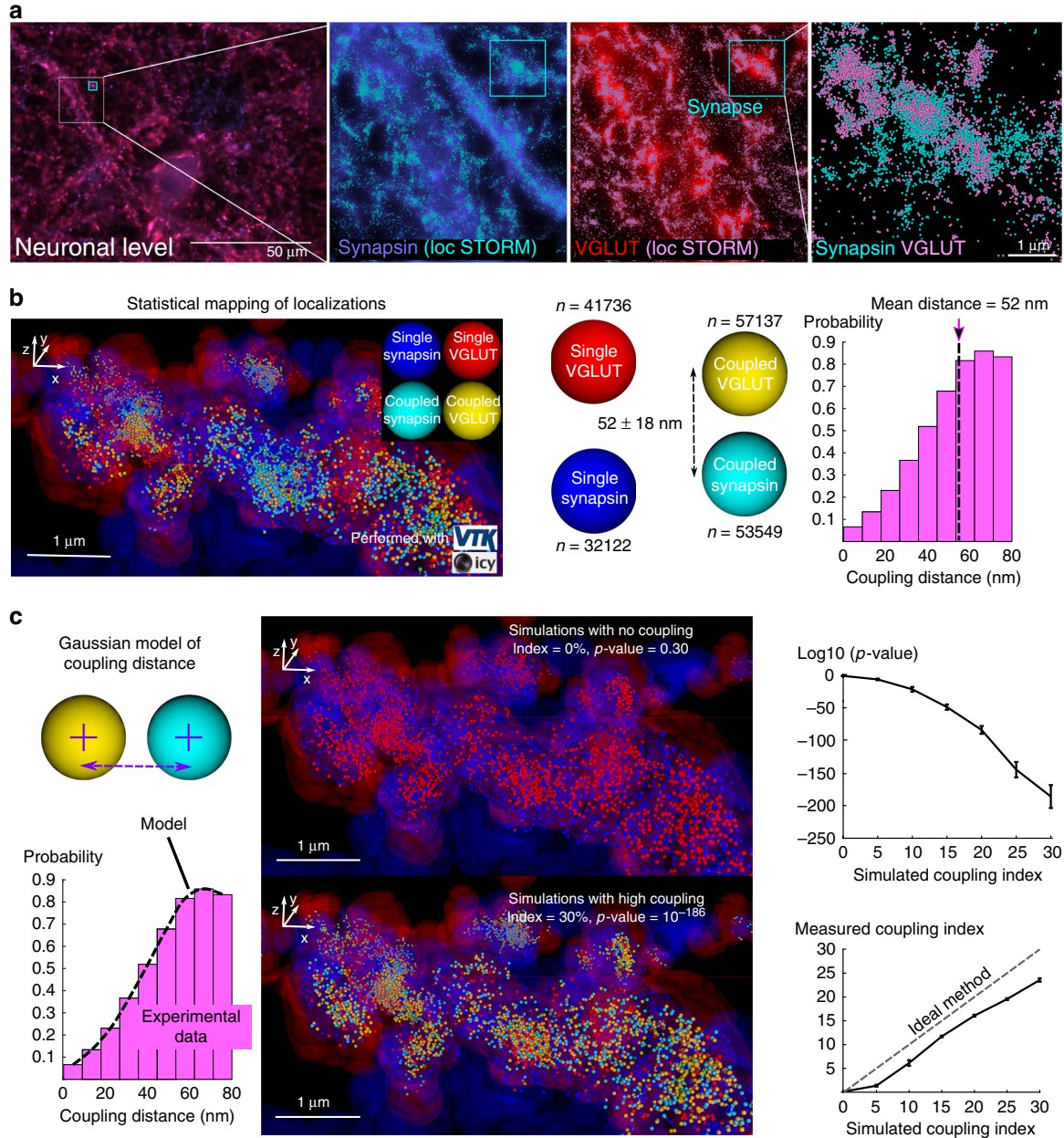

**Fig. 5** Combining SODA and 3D-STORM imaging to map the coupling between VGLUT and Synapsin localisations inside presynaptic boutons. **a** VGLUT and Synapsin are imaged in cultured hippocampal neurons of mice with 3D-STORM. Field of view is a 20 microns square, with 2 microns depth. STORM localisations of Synapsin (cyan) and VGLUT (magenta) are super-imposed to wide-field channels on Bruker Vutara Srx software (see also Supplementary Movie 1). High densities of molecules' localisations correlate with bright areas in wide field microscopy. Inside synapses, the 3D localisations of Synapsin and VGLUT are densely packed together. **b** Using DBSCAN method (Material and Methods), volumes with densely packed VGLUT (red enveloppe) and Synapsin (blue enveloppe) are automatically delineated. The intersection between VGLUT and Synapsin volumes corresponds to putative synaptic boutons. Inside putative boutons, single (VGLUT: red localisations, Synapsin: blue localisations) and coupled (VGLUT: yellow localisations, Synapsin: cyan localisations) are statistically mapped with SODA. 3D VTK rendering in Icy is used to visualise localisations (coloured spheres). Histogram of computed coupling distances with SODA is shown (mean = 52 nm, $n \approx 250{,}000$ individual couples, s.e.m = 0.04 nm). **c** The coupling distance is modelled with a (thresholded) Gaussian process with mean = 68 nm, s.d. = 28 nm and upper-bound = 80 nm (Material and Methods). Synapsin localisations are simulated around experimental VGLUT localisations inside the extracted putative boutons (ROIs), for increasing coupling index (0% (no coupling) to 30% (high coupling)). For each simulated coupling index, coupling parameters are estimated with SODA, and the single and coupled localisations are statistically mapped (error bars = ±s.e.m, N = 8 simulations per coupling index (i.e. 2 simulations per $n = 4$ STORM images)). Black dashed line corresponds to the ideal method that would estimate a coupling index equal to the simulated ground truth

neuron image containing about 5000 spots. For 3D STORM images with typically ~45,000 localisations, the computation time is in the order of few minutes (<10′) for the whole pipeline analysis. Combination of an easy-to-use graphical programming developed in Icy[17], with advanced VTK-based visualisation and

graphical rendering, and the overall low computation time, enables the straightforward use of SODA in batch analysis on very large data sets.

It is worth noting that, as SODA uses objects' localisations, it applies to any type of fluorescence microscopy, ranging from

wide-field to localisation-based microscopy (PALM, STORM), through SIM. Moreover, the spatial resolution of the mapping is only limited by the localisation precision which amounts to just a few nanometres even for wide-field microscopy[58]. Thus, SODA makes the in cellulo multi-colour mapping of molecular assemblies with high spatial resolution easy and automatic, and could favourably complement the use of electron microscopy, which is the gold standard in terms of resolution resolution but can be time consuming when doing robust statistical measurement analysis on multiple labelling assays.

## Methods

**Computation of $G^0$.** The Ripley-based vector $\mathbf{G}$ is the sum of a normal vector $\mathcal{N}(\boldsymbol{\mu}, \boldsymbol{\sigma})$ that counts the number of random (red) objects in each ring, and of a coupled vector $\mathbf{C}$ that counts the additional number of coupled objects. The component $C_i$ of $\mathbf{C}$ counts the total number of (red) objects that are coupled to (green) objects inside $\mathrm{Ring}(r_i, r_{i+1})$. We highlight that $C_i$ is an overestimate of the number of (red) $A_2$ objects with a coupling distance comprised between $r_i$ and $r_{i+1}$. Indeed, when (green) $A_1$ objects are densely packed, other (red) objects with different coupling distances that are coupled to (green) neighbors can also lay within $\mathrm{Ring}(r_i, r_{i+1})$. We can thus decompose $C_i = \widetilde{C}_i + \sum_{j \neq i} \alpha_{i,j} \widetilde{C}_j$, with $\widetilde{C}_i$ the exact number of couples with a coupling distance comprised between $r_i$ and $r_{i+1}$. The weighted sum $\sum_{j \neq i} \alpha_{i,j} \widetilde{C}_j$ counts the total number of (red) coupled objects with coupling distance comprised between $r_j$ and $r_{j+1}$ with $j \neq i$ but that lay within $\mathrm{Ring}(r_i, r_{i+1})$ around some (green) objects. We highlight that this weighted contribution tends to 0 when (green) objects are well separated and do not share (red) couples. The weight $\alpha_{i,j}$ is equal to the proportion of $\mathrm{Rings}(r_i, r_{i+1})$ that overlap with $\mathrm{Rings}(r_j, r_{j+1})$ around (green) $A_1$ objects. In a matrix form, the coupling decomposition reads $\mathbf{C} = \mathbf{A}.\widetilde{\mathbf{C}}$ with $\mathbf{A}[i, i] = 1$ and $\mathbf{A}[i, j \neq i] = \alpha_{i,j}$.

To statistically estimate each component of the coupling vector $\widetilde{\mathbf{C}}$, we use the reduced Ripley's vector $\mathbf{G^0} = \frac{1}{\sigma}\mathbf{A}^{-1}.[\mathbf{G} - \boldsymbol{\mu}]$. Indeed, under the null hypothesis of (red) objects' randomness, $\mathbf{G^0}$ is a Gaussian vector with zero mean and unit variance. Moreover $\sigma \mathbf{G^0}$ is proportional to the number of couples $\widetilde{\mathbf{C}}$ at different distances, and thereof, it can be used to compute the coupling probability for each individual pair of objects.

**Estimation of the coupling probability $P(x, y)$.** Each component $G_i$ of the Ripley-based vector $\mathbf{G}$ is proportional to the number of (red) objects $A_2$ that lay within $\mathrm{Ring}(r_i, r_{i+1})$ around (green) objects $A_1$ (Table 2), and the total number of (red) objects within $\mathrm{Ring}(r_i, r_{i+1})$ is given by

$$\text{Total Number of (red) objects in Ring}(\mathrm{r}_i, \mathrm{r}_{i+1}) = \frac{n_1 n_2}{\text{Volume of the ROI}} G_i. \quad (4)$$

On the other hand, the number $\widetilde{C}_i$ of (red) coupled objects inside $\mathrm{Ring}(r_i, r_{i+1})$, after correction of rings' overlap, is given by

$$\widetilde{C}_i = \frac{n_1 n_2 \sigma_i G_i^0}{\text{Volume of the ROI}} \mathbf{1}\{G_i^0 > T(N)\}. \quad (5)$$

Thus, the coupling probability $P(\mathbf{x}, \mathbf{y})$ between a (green) object located at position $\mathbf{x}$ and a (red) object located at position $\mathbf{y}$ is equal to

$$P(\mathbf{x}, \mathbf{y}) = \sum_{i=0}^{N-1} \mathbf{1}\{r_i < d(\mathbf{x}, \mathbf{y}) \le r_{i+1}\} \times \frac{\widetilde{C}_i}{\text{Total Number of (red) objects in Ring}(\mathrm{r}_i, \mathrm{r}_{i+1})}$$
$$= \sum_{i=0}^{N-1} \mathbf{1}\{r_i < d(\mathbf{x}, \mathbf{y}) \le r_{i+1}\} \frac{\sigma_i G_i^0 \mathbf{1}\{G_i^0 > T(N)\}}{G_i}. \quad (6)$$

**Statistical test of spot coupling.** To build a statistical test of objects' coupling, we use the reduced vector $\mathbf{G^0}$ and use the maximal component $G_{\max}^0 = \sup_{1 \le i \le N-1} G_i^0$ to test statistically whether (red) objects are randomly distributed (null hypothesis), or if there is at least one ring where coupled objects accumulate significantly. For any $x > 0$, we have that

$$\Pr\{G_{\max}^0 \ge x\} = 1 - \Pr\{G_{\max}^0 < x\}$$
$$= 1 - \Pr\{\forall i, 0 \le i \le N-1, G_i^0 < x\}. \quad (7)$$

Because $G_i^0$, for $0 \le i \le N-1$, are independent normal variables, we have

$$\Pr\{\forall i, 0 \le i \le N-1, G_i^0 < x\} = (\Pr\{\mathcal{N}(0.1) < x\})^N. \quad (8)$$

that is

$$\Pr\{\forall i, 0 \le j \le N-1, G_i^0 < x\} = \mathrm{cdf}^N(x), \quad (9)$$

where $\mathrm{cdf}(x)$ is the cumulative density function of the standard normal law: $\mathrm{cdf}(x) = \int_{-\infty}^{x} \frac{1}{\sqrt{2\pi}} \exp^{-\frac{x^2}{2}} dx$. Finally, reinjecting Eq. (9) in Eq. (7), we obtain that

$$\Pr\{G_{\max}^0 \ge x\} = 1 - \mathrm{cdf}^N(x), \quad (10)$$

and the $p$-value is thus given by

$$p-\text{value} = 1 - \mathrm{cdf}^N(G_{\max}^0). \quad (11)$$

**Gaussian point process simulations.** In simulations, we use a Gaussian point processes where positions of coupled (red) points (=localisations) are distributed around (green) points. The radial coupling distance $r$, in turn, follows the absolute value of a normal law: $r = |u|$ with $u \sim \mathcal{N}(\mu_c, \sigma_c)$. For STORM imaging, we also add an upper-bound (threshold) for the radial distance $r < 80$ nm. The mean $\mu_c$ models the mean coupling distance between points and accounts for the type of coupling (direct interaction, synaptic apposition...) between molecules (Fig. 1). The s.d. $\sigma_c$, in turn, accounts for both the possible variations in the interaction distance due to thermal noise or organelle size for example, and the localisation' uncertainty.

**Generation of synthetic images.** We use a Mixed Poisson-Gaussian model to generate synthetic fluorescent images with size $256 \times 256$ pixels and a number $n_1 = n_2 = 100$ of fluorescent spots (chapter 1 of ref. 59). In this model, the intensity $I[x, y]$ at pixel location $[x, y]$ is equal to $I[x, y] = \text{gain} * U[x, y] + N(x, y)$ where $U$ is a random Poisson variable and $N$ an additive white Gaussian noise with mean 0 and s.d. equal to $\sigma_N$. The mean $\lambda[x, y]$ of the Poisson variable $U$ varies spatially: $\lambda[x, y] = P[x, y] + B$, $P[x, y]$ being the sum of the intensity of the particles generated in $[x, y]$ and $B = 50$ a constant background value. $\text{gain} = 1$ is the gain of the acquisition system. Finally, we assume an additive model for the intensity of the particles: $P[x, y] = \sum_{i=1}^{N} P_i[x, y]$, where $P_i[x, y]$ is the signal originating from the $i^{th}$ particle in pixel $[x, y]$. When a particle is significantly smaller than the resolution of the microscope, its intensity profile $P_i$ is well represented by the Gaussian PSF of the microscope[60] with a specific amplitude $A_i$: $P_i[x, y] = A_i e^{-\frac{(x - x_i^0)^2 + (y - y_i^0)^2}{2\sigma_{xy}^2}}$ where $[x_i^0, y_i^0]$ is the coordinate of the $i^{th}$ particle and $\sigma_{xy}$ the s.d. of the 2D Gaussian profile of the PSF. Particle amplitude $A_i$ was fixed to $A_i = 100$ for each particle $1 \le i \le N$. Finally, the s.d. $\sigma_N$ of the white Gaussian noise is computed based on the targeted SNR value[59]: $SNR = \frac{A_i}{A_i + B + \sigma_N^2}$, leading to $\sigma_N = \sqrt{\frac{A_i^2}{SNR^2} - (A_i + B)}$.

**Automatic segmentation of putative synaptic boutons.** To delineate automatically volumes where VGLUT and Synapsin are densely packed (and thus ignore isolated, background localisations), we adapt the Density-Based Spatial Clustering of Applications with Noise (DBSCAN) method[52], and implemented it in Icy. We thus define an ensemble of core points that have more than $m = 10$ neighbors at a distance below $d = 200$ nm. $m$ and $d$ are user-defined parameters, and we checked that the ensemble of core points was not affected too much by variations of these parameters. Then, we build the ROIs around densely packed localisations by taking the union of balls centred at computed core points with radius $d + \delta$, where $\delta$ is a dilatation parameter that smoothen the ROI's boundaries ($d = \delta$ here). Finally, the ROI corresponding to putative synaptic boutons and used to perform the coupling analysis with SODA is equal to the boolean intersection of VGLUT and Synapsin ROIs.

**Icy protocols.** We perform multi-steps batch analysis of SIM and STORM images using graphical programming plugin Protocols in Icy. Tutorial for Icy installation is provided (Supplementary Movie 2). Protocols' screenshots are shown in Supplementary Figs 3 and 4, and are publicly available on Icy website (http://icy. bioimageanalysis.org/protocol/list). Protocols' tutorials are provided as Supplementary Movies 3 and 4. Each protocol consists of multiple elementary blocks that perform sequential steps of the image analysis.

SIM protocol (Supplementary Fig. 3 and Supplementary Movie 3 (tutorial)): First, user specifies the folder that contains multichannel SIM images, and defines inputs of the protocol such as the channels of pre- and postsynaptic molecules, the channel used to define the dendritic mask, the scales and thresholds used in the wavelet detection of molecule spots[9] or the maximal search distance of SODA. Then a first series of blocks in the protocol delineate the dendritic mask (block 6, can be expanded by double-clicking on the block title Cell Mask). It consists of HK-means thresholding of the fluorescence intensity of the pre-defined image channels to segment the whole neuron and the brighter cell soma. Dendritic mask is then obtained by removing the soma mask to the neuronal mask. The mask is then dilated to cover also presynaptic Synapsin spots in adjacent axons. The second main step of SODA protocol consists of wavelet detection blocks (blocks 10-14-19)[9] to extract pre- and postsynaptic spots inside the dendritic mask. The next three blocks (16-20-24) of the protocol are the specific SODA block that statistically analyse the coupling between PSD95 and Homer, PSD95 and Synapsin, and Homer

and Synapsin respectively. Based on SODA analysis, we designed a specific block (32) named Trio to extract single isolated spots, couples and triplets, with their individual morphologies and the associated coupling probabilities and distances. Triplets are defined as ensembles of three different spots (Homer, PSD95 and Synapsin) with at least two strictly positive coupling probabilities among the three. An option in the block Trio "Select strict triplets" allows to select only triplets with three positive coupling probabilities (in our experiment, there are 3129 strict triplets, and 7930 triplets with at least two positive coupling probabilities).

STORM protocol (Supplementary Fig. 4 Supplementary Movie 4 (tutorial)): A first series of blocks (blocks 5–7) extract localisations' coordinates from microscope files. Then, a second series of blocks delineate three-dimensional ROIs around dense clusters of VGLUT and Synapsin localisations using the DBSCAN method (blocks 6–8). The intersection between VGLUT and Synapsin ROIs (block 9) is, with single-molecule localisations, an input of the SODA STORM 3D block (10) that statistically computes all the coupling probabilities between individual VGLUT and Synapsin localisations. Finally, a last series of blocks export SODA results (coupling parameters) in files (block 11) and map single and coupled localisations in 3D with VTK (http://www.vtk.org) in Icy (blocks 12-13-14 and 15).

**Generation of genome-edited cells and fluorescence staining**. Hep2, a Human cervical adenocarcinoma cell line (Clone 2B, misidentified BioSample: SAMN03151705) was a gift of A Dautry and was the parental cell line used to obtain the clone Hep2$\beta$, stably expressing *IL-2R$\beta$* gene as described in (Grassart et al. EMBO R, 2008). This cell line did not have any mycoplasma contamination as verified by the kit MycoAlert from Lonza. Hep2$\beta$ cells were edited for CLTA similarly to clathrin-GFP edited cells[61]: Briefly, Zinc-Finger-Nucleases (ZFNs) and donor plasmids were transfected into cells using a single cuvette Amaxa Nucleofector device (Lonza), as per the manufacturer's protocol, Nucleofector solution R and programme I-013. After transfection, cells were transferred to 37 °C, 5% CO$_2$. Recovered cells were sorted for GFP-positive signals using a DAKO-Cytomation MoFlo High Speed Sorter directly as single cells into 96-well plates. Cells were maintained under 5% CO2 at 37 °C in DMEM (Invitrogen) supplemented with 10% FBS (Biowest). Cells were grown on glass bottom dishes No 1.5 (MatTekTM) overnight and incubated the next day for 2 min at 37 °C with transferrin (Sigma-aldrich ref: T0665) coupled to Cy3 (house-made coupling with Fluorochrome CY3 monofunctional (GE Healthcare, Ref: Q13108)), or with house-made anti-IL2R 561[62] coupled to Cy3 (house-made coupling with Fluorochrome CY3 monofunctional (GE Healthcare, Ref: Q13108)) for 5 min. Then, cells were extensively washed and immediately fixed in 4% paraformaldehyde and 4% sucrose at room temperature for 20 min.

**Primary hippocampal neurons in culture**. Hippocampal cultures were obtained from 18-day-old mice (C57BL6N) or rat (Sprague Dawley) embryos. All male and female embryos were used and mixed per litter (usually from 6–8 for mice to 10–12 for rats). Similarly to ref. 63, hippocampi from E18 rodent embryos were dissociated by treatment with trypsin (0.25% for 15 min at 37 °C) followed by trituration with a constricted Pasteur pipette. The cells were plated onto poly-Ornithine-coated coverslips (1 mg/mL) (Sigma-Aldrich, St. Louis, MO) in 4-well tissue culture plates at density of $6 \times 10^4$ cells/well in MEM-HS medium (modified Eagles medium, 10% horse serum, 0.06% glucose, 100 units/mL penicillin, 100 µg/mL streptomycin, 500 µM Glutamax). After 1 h, when the cells were attached to the substrate, the medium was replaced with Neurobasal-B27 medium conditioned previously on confluent glial feeder layer (neurobasal medium (Gibco) containing 2% B27 supplement (Gibco), and 500 $\mu$M L-Glutamine (Sigma-Aldrich). Cultures were maintained 3 weeks at 37 °C in a humidified atmosphere of 95% air and 5% CO$_2$ to obtain mature hippocampal network and synapses. Neurobasal medium was conditioned overnight on a confluent astrocyte feeder layer. One third of the neuronal medium was then replaced with this fresh conditioned medium once a week.

**Immunohistochemistry**. Similarly to ref. 41, neurons were fixed with cold methanol for 5 min at −20 °C. Quenching with NH4Cl for 15 min was followed by a permeabilisation step for 4 min with a mixture of 0.1% Triton-X100/PBS/0.125% cold water fish skin gelatin (fish gelatin) (Sigma-Aldrich). After three PBS 1× washings, neurons were incubated in blocking solution containing PBS/0.25% fish gelatin for 30 min. Immunocytological staining was performed by incubation with the primary antibody in PBS/0.125% fish gelatin overnight at 4 °C, followed by an incubation in the secondary antibody in PBS/0.125% fish gelatin for 45 min at room temperature. The antibodies used were: guinea pig polyclonal antibody to Synapsin (synapsin 1/2 (#106004), dilution 2000e) and rabbit polyclonal anti-Homer (Homer 1(#160003), 200e) were from Synaptic Systems, mouse monoclonal anti-PSD95 (#P78352, 500e) was from NeuromAb, chicken polyclonal anti-MAP2 (#ab5392, 20000e) was from Abcam, and VGLUT1&2 antibody (#pab0047, 200e) was from Covalab[64].

**Total internal reflection fluorescence microscopy**. Total internal reflection fluorescence (TIRF) microscopy images were captured using Cell MTM software on an Olympus IX-81 microscope using a 100x/NA1.45 objective and an EMCCD camera IxonEM+ (Andor). A 488 nm solid-state laser (Olympus) and a 561 nm

solid-state laser (Olympus) were used to excite GFP and Cy3 fluorophores, respectively. Images were obtained without gain and an exposure time of 800 to 1000 ms. Simultaneous two colour TIRF images were obtained using a DV2 image splitter (Optical Insights) to separate GFP and Cy3 emission signals.

**Confocal microscopy**. Confocal images for synaptic triple labelling were obtained using a confocal microscope Leica TCS SP5 (Leica Microsystems CMS GmbH, Mannheim, Germany) and a ×63 objective (NA 1.4; followed by a digital zoom to achieve the ideal sampling). Images were acquired by sequential scanning of the emission lines. Alexa 488 was detected using the 488 nm-line of an argon laser for excitation; Cy3 and Cy5 were respectively excited by the 543 nm-line of a green neon laser and the 633 nm-line of a helium neon laser. Typically, sections (from 1024 up to 4096 pixels), were scanned three times, to optimise the signal/noise ratio.

**Structured-illumination microscopy**. Super-resolution structured-illumination microscopy (SIM) was performed on a Zeiss Elyra PS.1 system equipped with a 63x objective (N.A. 1.4) and an EMCCD Andor, iXon 885 camera. Three channels containing pictures (typically 1900 × 1900 pixels with a pixel size of 39 nm) were acquired with 4 lasers (405, 488, 561 and 642 nm) and five different grids.

Quantification was performed on 9 neurons for each condition which correspond roughly to 20,000 to 30,000 immunoreactive spots for each channels. Statistical significance was evaluated using Graphpad prism software. The level of significance (Mann-Whitney) is indicated by one ($p < 0.05$), two ($p < 0.01$), or three ($p < 0.001$) symbols.

**Stochastic optical reconstruction microscopy**. Stochastic optical reconstruction microscopy (STORM) imaging was performed on a Vutara microscope (Bruker) with a high-numerical aperture (NA) objective (60x, water, NA 1.2, Olympus). 170 nm coverslips (Menzel glaser 18 mm diameter #1.5) were mounted on a glass slide with a 15mm hole. The hole was filled with imaging buffer (Tris 50 mM, NaCl 10 mM, 10% glucose, 100 mM MEA, 70 U/mL glucose oxidase (Sigma G0543), 20 g/mL catalase) and sealed with Picodent twinsil. Samples were illuminated successively with a 647 and 488 nm laser and a 405 nm laser for the reactivation of the 488 fluorophores.

Neurons were isolated with wide field mosaïc microscopy (Cool snap camera) and then imaged for STORM for a series of 30 000 images with a FLASH4 CMOS camera (20 ms, 20 × 20 microns). 3D-STORM imaging was done using the bi-plane module allowing the localisation in the xyz direction. The Srx software (Bruker) was used to localise particles in 3D. Localisation tables were exported and used in ICY software for statistical localisation analysis.

**Correction for chromatic aberration**. TIRF images have been corrected using beads' alignment with rigid registration (plugin Rigid registration in Icy) and then, (inverse) rigid transformation of images.

Correction for chromatic aberration in super-resolution microscopy (SIM and STORM) has been done using multispectral (blue/green/orange/dark red) Tetraspeck beads (Thermofisher T7279). Channel alignment has been done on SIM in Zeiss software or on STORM in Bruker's software.

**Code availability**. Code (Icy protocols and plugins) can be freely dowloaded from Icy website (http://icy.bioimageanalysis.org/list) or directly within Icy software through the search bar.

**Data availability**. All data generated or analysed during this study are included in this published article (and its supplementary information files), or are available from the authors on reasonable request.

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

## Acknowledgements

We deeply thank Dr. Carl G. Ebeling (Worldwide Application Scientist for Bruker Fluorescence Microscopy), and Jeff Stuckey (Product Marketing Manager, Bruker) for their help and the use of the Bruker Vutara system and its SRX visualisation and analysis software for rendering localisation data. We acknowledge the ImagoSeine core facility of the Institut Jacques Monod, member of IBiSA and the France-BioImaging infrastructure, supported by ANR grant (ANR-10-INBS-04). We thank the NeurImag imaging facility of the Centre for Psychiatry and Neuroscience, the labex 'Who am I' and the national research group in Microscopy of the living (GDR ImaBio). This work was financially supported by the Institut Pasteur, the CNRS, the INSERM, ANR grant ANR-10-LABX-62-IBEID and a grant from the Institut pour la Recherche sur la Moelle épinière et l'Encéphale. T.L. was funded by a Bourse Roux from Institut Pasteur. A.G. is funded by a Bourse Roux from Institut Pasteur. Finally, we warmly thank Fabrice de Chaumont and Anna Raffaela Damato for their critical reading of the manuscript. The funders had no role in study design, data collection and analysis, decision to publish, or preparation of the manuscript.

## Author contributions

T.L. designed the method, performed the statistical analysis and simulations, analysed the imaging and synthetic data. A.G. and N.S. performed the TIRF imaging of endocytosis molecules (Fig. 2d). A.D designed the protocol plugin in Icy. S.D helped with the java programming in Icy. O.F. helped for the use of the Elyra microscope (three-colour SIM imaging). L.D. performed the super-resolution imaging (three-colour SIM and three-dimension STORM) of synaptic molecules, designed the analysis protocol for SIM images in Icy. T.L. and L.D. interpreted results. J.-C.O.-M. and L.D. supervised the study and provided overall guidance. T.L., L.D. and J.-C.O.-M. wrote the manuscript.

## Additional information

**Competing interests:** The authors declare no competing financial interests.

