## [Peer Review File · Nature Communications]

Reviewers' comments:

Reviewer #1 (Remarks to the Author):

The authors present an analysis framework based on marked point-process statistical analysis. This framework is applied to the problem of co-localization, and is demonstrated on simulated and real data (known and unknown).

Readability: the manuscript lacks accessibility due to the technical jargon and lack of clearly corresponding visuals.

I guess that Probability $\{(x; y) \text{ is a Couple}\}$ is $P(x,y)$. There are a number of other quantities which are not directly defined. A list of variables and definitions may help.

What is point-process? Marked point-process? It is important to define these in plain language, so that the general audience can understand the method.

How is this different than the pair cross-correlation function? It also normalizes by the average density. Thus, gives a reference value (is 1 when random, >1 when correlated and <1 when anti-correlated). I guess the difference is related to the generation of a "probability" that two objects are actually coupled. Is this really not an average property based on all such pairs of objects with the same distance?

synaptic buttons = synaptic boutons (yes, we took the French word).

What does the coupling distance really mean (quantitatively). For example , 100 nm for the Tf and CCP? Is it the average distance? Weighted mean?

Does this method solve the asymmetry issue of Pearson? That is, the correlation one gets depends on which molecule you choose as the reference?

There are other point-based methods that should be mentioned and compared with SODA. For example, Heilemann (Malkusch et al, Histochem Cell Biol 2012), Teraska (Larson et al, MBoC 2014) methods.

Reviewer #2 (Remarks to the Author):

In this article Authors describe SODA, a statistical method to characterize spatial positioning of molecules using MPP (marked point processes). Most importantly, the method is insensitive to noise and the choice of PSF. The topic is timely and very relevant with the recent advancements in high-resolution microscopy.

My worry with MMPs and other energy minimization methods is the they are computationally intensive, but Authors report reasonable runtime that will be affordable for most applications.

The method mathematically sounds and seems theoretically well grounded.

Real tests show the great capabilities both in 2 and 3D.

Software provided.

A great summary of the state of the art is given and I really appreciate supplementary table 1. Probably it would be worth moving it to the main text or write a small review article separately.

In general I do not have any major criticism. This is one of those few papers that meet my general

expectations both in terms of presentation quality and the importance of the content.

I support its publication.

Reviewer #3 (Remarks to the Author):

The study presented here by Olivo-Marín and collaborators is a highly developed method of analyzing the correlation (and/or colocalization) of various image elements, in multiple colors. The authors present their method in thorough detail, and compare it with many standard current tools, using simulated data, as well as some real images. The results are convincing.

The authors then showcase their analysis relying first on structured illumination. Here they describe differences in the localization of synaptic proteins, which are entirely plausible. The analysis is then applied to STORM images, where the potential colocalization of two synaptic vesicle proteins is analyzed. The results are again plausible, albeit more difficult to interpret.

I find the study overall excellent. I only have two minor comments:

1) The authors should consider adding to their discussion a short section indicating that imaging artifacts, such as problems with labeling, for example caused by poorly tested antibodies, will affect the results and the interpretations. For example, the 5:5 stoichiometry they find for synapsin and the glutamate transporter is probably best explained by the antibodies not having labeled all of the copies of these molecules on the vesicles. Both are expected to have higher copy numbers, according to biochemical results (Takamori et al., *Cell*, 2006; Wilhelm et al., *Science*, 2014).

2) Will the authors make the software available via a form of plug-in on a widely used platform, such as ImageJ? This would be important to mention, even if it is only planned to do so, and not yet realized.

Reviewers' comments:

Reviewer #1 (Remarks to the Author):

The authors present an analysis framework based on marked point-process statistical analysis. This framework is applied to the problem of co-localization, and is demonstrated on simulated and real data (known and unknown).

Readability: the manuscript lacks accessibility due to the technical jargon and lack of clearly corresponding visuals.

I guess that Probability $\{(x; y) \text{ is a Couple}\}$ is $P(x,y)$. There are a number of other quantities which are not directly defined. A list of variables and definitions may help.

Answer: We have simplified the section I.A. « SODA » of the manuscript. We have clearly defined all the mathematical notations and moved some equations to a new subsection « Spatial Analysis/ Estimation of the coupling probability $P(x,y)$ » in the Material and Methods section. We have also listed all the mathematical notations and expressions with their meaning in a new table (Table I, « Mathematical variables »).

What is point-process? Marked point-process? It is important to define these in plain language, so that the general audience can understand the method.

Answer: We have changed the first paragraph of the results section IA, and included the following sentences:

« To statistically characterise the spatial distribution of molecules inside cells, we model the positions of single or aggregated molecules with a Marked-Point-Process [P. J. Diggle, Statistical analysis of spatial and spatio-temporal point patterns (CRC Press, 2013)], where the Mark is the ensemble of attributes of each individual fluorescent spot (color, size, shape..., see fig. Fig. 1-d-1), and the Point-Process is a mathematical model where the localizations of spots are viewed as a collection of points randomly located inside the cellular region of interest (ROI). Point-processes are powerful statistical tools for modeling and analysing spatial data that have demonstrated their strength in such diverse disciplines as forestry [P. Haase, Journal of Vegetation Science 6, 575 (1995)], cell biology [E. Diaz et al., IEEE Trans. on PAMI 30, 1659 (2008)] or neurosciences [E. N. Brown, R. E. Kass, and P. P. Mitra, Nat Neurosci 7, 456 (2004)]. To characterise the spatial relations between two populations A1 (green) and A2 (red) of objects (spots or localizations), we use the Ripley's K function [27], a gold standard for analysing the second-order properties (i.e. distance to neighbors) of point-processes. »

How is this different than the pair cross-correlation function? It also normalizes by the average density. Thus, gives a reference value (is 1 when random, >1 when correlated and <1 when anti-correlated).

Answer: This is a very good point : actually the Ripley-based function G that we introduce is a discrete, pair-correlation function that contains an additional boundary term that corrects the under-counting of object's neighbours near the boundary of the region of interest (ROI). We have added the following sentence at the end of the first paragraph of section I.A.:

« For a given distance parameter r , the $K(r)$ function counts all the pair of objects closer than r ,

and it is therefore difficult to precisely extract the distances where coupled objects accumulate. To count the number of (red) objects at specific distance from (green) objects, we introduce the function $G = [K(r_{i+1}) - K(r_i)]_{i=0..N-1}$, composed by incremental subtractions of the K function for a series of increasing concentric distances $r_0 = 0 < r_1 < \dots < r_N$ (Fig. 1-d-2). G is actually the (discrete) pair-correlation function with a boundary correction term, and is proportional to the number of A_2 (red) objects that fall inside the different Ring(r_i, r_{i+1}) around A_1 (green) objects (see Table I for the detailed definitions of variables and expression of G). »

I guess the difference is related to the generation of a “probability” that two objects are actually coupled.

Answer: Our main contribution is indeed the complete characterisation of the probability law of the « pair-correlation » function G under the null hypothesis of objects’ spatial randomness. This, in turn, allows us to compute the coupling probability between all the pairs of objects.

Is this really not an average property based on all such pairs of objects with the same distance?

Answer: Yes it is: the computation of individual coupling probabilities is actually an average property based on the statistical accumulation of coupled (red) A_2 objects in a subset of rings around (green) A_1 objects. The estimation of accumulated objects is based on: (i) the characterisation of the probability law of the G function and (ii) the estimation of the spatial interactions between densely-packed A_1 objects. This estimation is summarized in the interaction matrix A (see table I) and takes into account the exact positions of A_1 objects.

synaptic buttons = synaptic boutons (yes, we took the French word).

Answer: We are happy to see that French is still used in science! We have changed buttons to boutons and checked the grammar throughout the manuscript.

What does the coupling distance really mean (quantitatively). For example , 100 nm for the Tf and CCP? Is it the average distance? Weighted mean?

Answer: The coupling distance is the probability-weighted mean of the distances between all the pairs of objects (Equation (3) in the revised version of the manuscript).

Does this method solve the asymmetry issue of Pearson? That is, the correlation one gets depends on which molecule you choose as the reference?

Answer: It does solve it and it also provides information on (possible) different stoichiometry. Coupling index can be evaluated for both molecules.

There are other point-based methods that should be mentioned and compared with SODA. For example, Heilemann (Malkusch et al, Histochem Cell Biol 2012),

Answer: This study, as well as the study of Owen et al published two years later in the same journal (Histochem Cell Biol 141, 605 (2014)), use second order properties (Nearest neighbours for Malkush et al. and Ripley’s K function for Owen et al.) to define and segment the dense clusters of localizations in the different colours/channels. On that basis the two studies use standard correlation coefficients (Spearman coefficient in Malkusch et al, and

Pearson coefficient in Owen et al) to measure the co-localization between clusters. These methods present the advantage of being adapted to localization-based microscopy, but (i) they do not provide the statistical significance of the computed co-localization coefficient, (ii) they are sensitive to the coupling distance between localizations and, (iii) they do not provide the coupling probabilities of the individual pairs of localisations.

We thank the referee for having pointed out this class of localisation-based co-localization methods, that we have listed in the summary table II and discussed in the first paragraph of section D.3:

«Different co-localisation methods have been proposed in localisation-based, super-resolution microscopy [Malkusch et al, Histochem Cell Biol (2012), Rossy et al. Histochem Cell Biol (2014)]. These methods delineate localisation clusters with second-order spatial analysis (nearest-neighbor or Ripley K function), before measuring the overlap between the clusters in different colours with standard correlation coefficients (Spearman or Pearson). While these methods have the advantage that they can be applied to single localisations, the empirical computation of the statistical significance of each correlation coefficient would need multiple randomisations of the tens of thousands single localisations in each image. Moreover, these methods do not measure the coupling properties (probability and distance) between individual pairs of localisations (see Table II). »

Teraska (Larson et al, MBoC 2014) methods.

Answer: This method uses the Point-Process framework to characterise the spatial distribution of the different endocytic molecules at the cell membrane, not their colocalisation that is measured with (local) Pearson coefficients. We have added this reference in the introduction and in Table II that summarises the different co-localisations methods.

Reviewer #2 (Remarks to the Author):

In this article Authors describe SODA, a statistical method to characterize spatial positioning of molecules using MPP (marked point processes). Most importantly, the method is insensitive to noise and the choice of PSF. The topic is timely and very relevant with the recent advancements in high-resolution microscopy.

My worry with MMPs and other energy minimization methods is the they are computationally intensive, but Authors report reasonable runtime that will be affordable for most applications.

The method mathematically sounds and seems theoretically well grounded.

Real tests show the great capabilities both in 2 and 3D.

Software provided.

A great summary of the state of the art is given and I really appreciate supplementary table 1. Probably it would be worth moving it to the main text or write a small review article separately.

In general I do not have any major criticism. This is one of those few papers that meet my general expectations both in terms of presentation quality and the importance of the content.

I support its publication.

Answer: We thank the reviewer for this positive appreciation of our work.

Reviewer #3 (Remarks to the Author):

The study presented here by Olivo-Marin and collaborators is a highly developed method of analyzing the correlation (and/or colocalization) of various image elements, in multiple colors. The authors present their method in thorough detail, and compare it with many standard current tools, using simulated data, as well as some real images. The results are convincing. The authors then showcase their analysis relying first on structured illumination. Here they describe differences in the localization of synaptic proteins, which are entirely plausible. The analysis is then applied to STORM images, where the potential colocalization of two synaptic vesicle proteins is analyzed. The results are again plausible, albeit more difficult to interpret.

I find the study overall excellent. I only have two minor comments:

1) The authors should consider adding to their discussion a short section indicating that imaging artifacts, such as problems with labeling, for example caused by poorly tested antibodies, will affect the results and the interpretations. For example, the 5:5 stoichiometry they find for synapsin and the glutamate transporter is probably best explained by the antibodies not having labeled all of the copies of these molecules on the vesicles. Both are expected to have higher copy numbers, according to biochemical results (Takamori et al., Cell, 2006; Wilhelm et al., Science, 2014).

Answer: We thank the referee for having pointed out the study of Wilhelm et al. that we did not mention in the previous version of our manuscript. We have now completed the discussion at the end of section D.3:

«Moreover, the measured high coupling stoichiometry is in accordance with the average 8 Synapsin and 9 VGLUT1 molecules per synaptic vesicles that has been previously reported with purification and mass-spectrometry analysis [Takamori et al. Cell (2006), Wilhelm et al. Science (2014)]. The slightly lower stoichiometry that we report might be due either to unlabeled molecules, or to the fact that previous proteomic analysis were either performed on entire brains [Takamori et al. Cell (2006)] or on the cerebellum and cortex [Wilhelm et al. Science (2014)] of adult rat brains, while we focus here on the molecular organization of rodent hippocampi.»

It should be noted that stoichiometry estimated with SODA is directly dependent on the STORM labelling efficiency (antibody affinity, blinking efficiency, number of acquired images...). Thus, one should thus be careful and use high-affinity and well-characterised primary and/or secondary antibodies to get a robust estimation of stoichiometry.

2) Will the authors make the software available via a form of plug-in on a widely used platform, such as ImageJ? This would be important to mention, even if it is only planned to do so, and not yet realized.

Answer: We are very thankful to the reviewer to highlight the need to make SODA available through a widely used software platform. Although we agree that ImageJ is indeed a software that everyone knows, it however has a number of drawbacks that justified that we and others decided at somepoint to develop more powerful platforms (Icy, ImageJ2, Fidji, ...). Moreover, even if Icy is not as widely used as ImageJ, it is steadily gaining an increasing popularity (~400 citations of the related Nature Methods article (de Chaumont et al. (2012) and almost 2,000 regular users <http://icy.bioimageanalysis.org/index.php?display=icyStats>) in the quantitative bioimaging community.

The main advantages of implementing SODA on Icy are the following:

- (i) Its ease of use, the variety and power of the developed plugins, the possibility of visual programming (protocols), a web based distribution and direct contact with the developers through chats;
- (ii) ImageJ is embedded within Icy which makes it easy to users to run both software from a unique program and thus allows the inter-operability and passing of data/parameters between different plugins (Icy) and macros (ImageJ);
- (iii) Icy offers the possibility of developing protocols that perform automatically (batch processing) the multiple steps of complex image analysis pipeline (in the case of SODA, extraction of dendritic masks, automatic detection of fluorescent spots, SODA analysis, export of coupling properties (probability, distance) and statistical mapping of individual couples). Such pipeline would be extremely tedious to export in other imaging softwares;
- (iv) Icy includes a high-end 3D rendering powered by VTK (<https://www.vtk.org>), that makes it possible to render and visualize the mapping and coupling between single localizations in 3D STORM in a very efficient and informative manner.

As a complement, and to further increase the user-friendliness of Icy, we propose 3 demonstration videos explaining how to Install Icy (Supplementary video 2) and run either the protocol for analysing SIM images (Supplementary video 3) or 3D-STORM localisations (Supplementary video 4).